# Effect of a Mineral–Microbial Deodorizing Preparation on the Functions of Internal Organs and the Immune System in Commercial Poultry

**DOI:** 10.3390/ani11092592

**Published:** 2021-09-03

**Authors:** Joanna Kowalczyk, Bartłomiej Tykałowski, Marcin Śmiałek, Tomasz Stenzel, Daria Dziewulska, Andrzej Koncicki

**Affiliations:** Department of Poultry Diseases, Faculty of Veterinary Medicine, University of Warmia and Mazury, 10-719 Olsztyn, Poland; bartlomiej.tykalowski@uwm.edu.pl (B.T.); marcin.smialek@uwm.edu.pl (M.Ś.); tomasz.stenzel@uwm.edu.pl (T.S.); daria.pestka@uwm.edu.pl (D.D.); koncicki@uwm.edu.pl (A.K.)

**Keywords:** ammonia, odours, poultry production, deodorizing preparation, immune system

## Abstract

**Simple Summary:**

Poultry production generates the largest volumes of atmospheric ammonia and greenhouse gases such as methane, nitrogen oxide, and hydrogen sulfide. These gases have a negative impact on the health of living humans and animals. In our study, we evaluated the influence of one deodorizing biopreparation on the functions of organs and the immune system in poultry. The obtained results show no effect on a preparation on the physiological status of chickens and turkeys, although the improvement of housing conditions and reducing gas emissions which was confirmed by other authors.

**Abstract:**

Animal production is identified as one of the main sources of high concentrations of odours, which are related to air pollution, health problems of living organisms and indirect negative impact on production results. One common method for reducing emissions of ammonia is using preparations containing probiotics and hygroscopic or disinfecting compounds. This study was undertaken in order to determine the impact of innovative mineral–microbial deodorizing preparation, which reduces odorous gases, applying to the litter once a week in poultry houses on the physiological status of breeder chickens, broiler chickens and turkeys. Samples were collected after slaughter and analyzed using ELISA tests, flow cytometry and biochemical methods. Biochemical markers of the liver and kidney profile (ALT, AST, LDH, ALP, CK, TP, CALC, PHOS) and the titers of specific antibodies against AEV, aMPV, AAvV-1, IBDV, HEV, BA were analyzed in serum samples. The percentage contribution of T and B lymphocyte subpopulations was determined in the samples of tracheal mucosa, blood, and spleen. No significant differences were found between the control and experimental group with regard to all the analyzed parameters, with some exceptions for biochemistry. The results of our study indicated that mineral–microbial deodorizing preparation did not affect the physiological status of birds.

## 1. Introduction

Agriculture and animal production are perceived as the greatest anthropogenic sources of air contamination. When coupled with the pressure to protect the environment and act more ecologically, this perception of these industries urges producers to seek, develop, and deploy ever newer methods of reducing emissions of harmful gases. Animal production, including poultry production, generates the largest volumes of atmospheric ammonia (NH_3_) and greenhouse gases such as methane (CH_4_), nitrogen oxide (N_2_O), and hydrogen sulfide (H_2_S) [1,2]. Ammonia mainly derives from nitrogen present in animal feces, and the feasibility of its reduction is directly determined by the species, number, and genetic potential of animals, their feeding and maintenance strategies, and manure management [3,4,5]. Projects and actions aimed at reducing gas emissions can be implemented as early as during the planning of the production process. These may include designing facilities to employ easy-to-clean materials in their construction, providing appropriate microclimatic conditions, heating, and ventilating facilities with highly efficient systems, equipping buildings with automatic systems for removal of droppings and manure and installing exhaust air-purifying elements [6,7,8].

Apart from polluting atmospheric air and affecting the health of living organisms including man adversely, gases emitted from animal production also have an indirect negative impact on production results. Numerous studies have shown that high ammonia levels negatively affect feed conversion, final body weight of birds, carcass quality, laying production, length of the production cycle, and the incidence of opportunistic infections and respiratory diseases in poultry [9,10,11]. As early as 1978, Oyetunde et al. [12] demonstrated that ammonia and dust present in the air induce macroscopic and microscopic lesions in the trachea, lungs, and air sacs, thus debilitating defense systems and increasing the incidence of infections with *Escherichia coli* and other pathogens. In addition, high concentrations of harmful gases have been shown to negatively affect nervous and cardiovascular systems, acid–base homeostasis, and bird behavior, and to be largely determined by the bird housing system [13]. Studies have also shown that the litter system generates 10-fold higher NH_3_ emissions and 45-fold higher H_2_S emissions to the environment compared to the cage system [13,14].

The available literature describes multiple viable techniques for reducing emissions of ammonia from poultry production to the natural environment. They can be generally divided into feed-related (reducing protein intake and using feed additives and probiotics), husbandry-related (reducing stock density, controlling the microclimate, and providing good-quality litter), chemical (using iron vitriol (FeSO_4_·7H_2_O) or other compounds), and biological and physical solutions (installing air filters and biofilters, drying fecal matter, and deodorizing litter with agents for the purpose) [6,7,8,15,16]. One common practice to reduce gas emissions from buildings where birds are kept is the regular spraying of the litter with preparations containing probiotics and hygroscopic or disinfecting compounds. Probiotic preparations contain live microorganisms intended to saturate the environment with beneficial microflora and thus reduce emissions of ammonia and other odorous gases. Other chemical means of litter remediation include compounds of silicon, aluminum, and chlorine, characterized by the ability to disinfect waste or to bind water and heavy metals. These activities effectively inhibit the multiplication and development of microorganisms that metabolize nitrogen compounds, thus contributing to the reduction of emissions of this gas to the atmosphere [6,7,8].

Previous studies and analyses of animal husbandry parameters have shown the mineral–microbial deodorant Deodoric (University of Technology, Lodz, Poland) to be effective in reducing the concentration of odorous gases [10,17]. This background suggested the undertaking of a study to determine the impact of applying Deodoric to litter in poultry houses on the functioning of avian internal organs and the immune system. The investigation encompassed both its direct impact on the bird’s body and its indirect effect caused by the improvement of housing conditions.

## 2. Materials and Methods

Bird handling and all experimental procedures were performed in accordance with the guidelines for Care and Use of Laboratory Animals of the Local Ethical Commission in Olsztyn, Poland.

Each experiment used the deodorizing preparation Deodoric^®^ composed of two fractions: a dried fraction, being a mixture of six highly active bacterial strains (*Pseudomonas fluorescens* (ŁOCK pure culture collection of the Institute of Fermentation and Microbiological Technology in Łódź, Poland (ŁOCK) 0961), *Enterococcus faecium* (ŁOCK 0965), *Bacillus subtilis* (ŁOCK 0962), *Bacillus megaterium* (ŁOCK 0963), *Leuconostoc mesenteroides* (ŁOCK 0964), and *Lactobacillus plantarum* (ŁOCK 0996)) spray-dried with trehalose (5% *w*/*v*) and maltodextrin (Maltodextrin N 15% *w*/*v*; DE = 7–13, HORTIMEX Sp. z o. o., Konin, Poland), and a mineral carrier (perlite and bentonite in a 15%: 85% weight ratio and a 1:1 volumetric ratio) [18].

### 2.1. Birds

Three independent experiments were conducted within the study using parent-stock chickens, broiler chickens, and broiler turkeys. The study was conducted in October 2016–June 2017.

#### 2.1.1. Parent-Stock Chickens

Seventeen-week-old ROSS-308 parent-stock chickens (120 hens and 12 roosters) were transported from a commercial breeding farm to the laboratory of the Department of Avian Diseases, Faculty of Veterinary Medicine of the University of Warmia and Mazury in Olsztyn, Poland, where they were kept throughout the study (145 days). The immunoprophylaxis program for infectious diseases implemented in this flock is presented in Table 1. The birds were divided into two groups: the control group and the experimental group each of 60 hens and 6 roosters. Rearing conditions in the experimental rooms were consistent with the obligatory recommended standards and similar to commercial farm conditions (wheat litter to which fresh material was added once a week, stock density 5.5 birds/m^2^, temperature 20 °C, and humidity 70%). The birds were fed restricted rations (170 g/day), as recommended in the “ROSS Parent Stock Management Manual and ROSS-308 Laying Hen Nutrition Specifications” (Aviagen, Huntsville, AL, USA) [19] twice daily and had ad libitum access to water [10]. In the experimental group, the Deodoric^®^ (University of Technology, Lodz, Poland) biopreparation was applied manually to litter once a week in an amount of 170 g/m^2^, which was consistent with earlier calculations made based on stock density and excreta volume [9,20]. After the experiment was completed, the birds were euthanized in a specialized abattoir of the Department of Commodity Science and Animal Improvement, Faculty of Animal Bioengineering, University of Warmia and Mazury in Olsztyn.

#### 2.1.2. Broiler Turkeys

Ten-week-old Hybrid Converter turkeys (100 birds) were transported from a commercial farm (Grelavi S.A., Olsztyn, Poland) to the experimental premises of the Department of Avian Diseases, Faculty of Veterinary Medicine of the University of Warmia and Mazury in Olsztyn, where they were kept throughout the experiment (88 days). The birds were divided into two groups: the control group and the experimental group, each of 50 birds. The experiment was performed in two separated turkey houses with an area of 20 m^2^ each. Birds were kept on shredded dry wheat straw litter (depth—20 cm) and were provided a balanced complete feed (composition: crude protein 19.20%, oils and crude fats 4.20%, crude fiber 3.50%, raw ash 5.70%, lysine 1.17%, methionine 0.44%, calcium 1.00%, phosphorus 0.57%, sodium 0.16%) and water ad libitum. Once a week, poultry houses was supplied with the same amount of fresh dry wheat straw. In the experimental group, Deodoric^®^ was applied to litter once a week in a dose of 180 g/m^2^ [17].

#### 2.1.3. Broiler Chickens

One-day-old Cobb-500 broiler chickens (200 birds) were purchased and placed in the experimental rooms of the Department of Avian Diseases, Faculty of Veterinary Medicine of the University of Warmia and Mazury in Olsztyn, where they were kept throughout the experiment (45 days). Immediately after placement, the birds were vaccinated against infectious bronchitis virus (IBV) using a live strain H-120 IBV (CEVA Animal Health, Libourne, France) applied individually to a conjunctival sac. Next, they were divided into two groups: the control group and the experimental group, each of 100 birds, and reared on shredded dry wheat straw litter in rooms with a computer-controlled microclimate (mean temperature 26 °C, humidity 52%). The birds’ water and feed mixture were available ad libitum. In the room with the experimental birds, the Deodoric preparation was applied in a dose of 170 g/m^2^ every 7 days.

### 2.2. Sample Collection

Samples of blood, trachea, and spleen were collected from birds during slaughter, on the termination of individual experiments.

For serological analyses (*n* = 23) and biochemical analyses (*n* = 10), blood was sampled into Vacutainer CAT test tubes with a clot activator (Becton Dickinson, Franklin Lakes, NJ, USA) from each group in the experiment. After cooling at 4 °C for 24 h, the test tubes with blood were centrifuged at 1500× *g* for 15 min in an Allegra X-15R centrifuge (Beckman Coulter, Indianapolis, IN, USA) to separate serum. Blood samples were also collected into test tubes with EDTA K_2_ anticoagulant (Becton Dickinson, Franklin Lakes, NJ, USA) for cytometric analyses.

Immediately after slaughter, the trachea and spleen were sampled from eight birds from each group to isolate mononuclear cells for cytometric analyses.

### 2.3. Biochemical Analyses

Serum samples collected from breeder chickens, broiler chickens, and broiler turkeys were analyzed for biochemical markers of the liver profile (activities of: alanine aminotransferase (ALT), aspartate aminotransferase (AST), lactic dehydrogenase (LDH), and alkaline phosphatase (ALP)), those of the kidney profile (activity of creatine kinase (CK), total protein (TP) concentration, and creatine enzyme (Crea enz) concentration), as well as total concentrations of calcium (CALC) and phosphorus (PHOS). All determinations were conducted using an ACCENT-200 automatic biochemical analyzer (Cormay, Warsaw, Poland).

### 2.4. Serological Analyses

The serum of breeder chickens was analyzed for the titers of specific antibodies against avian encaphalomyelitis virus (AEV), avian metapneumoviruses (aMPV), infectious bursal disease virus (IBDV), infectious bronchitis virus (IBV), and avian avulavirus (AAvV-1). In turn, serum samples from broiler chickens were analyzed for the titers of antibodies against IBV, and those of broiler turkeys for the titers of specific antibodies against aMPV, hemorrhagic enteritis adenovirus (HEV), AAvV-1, and *Bordetella avium* (BA).

Antibody titers were determined using commercial aMPV, IBV, IBD, and AAvV-1 (Idexx, Westbrook, ME, USA) and AEV, BA, and HEV ELISA kits (Synbiotics, San Diego, CA, USA) following the producers’ guidelines. Individual stages of the ELISA test were performed using an epMotion automatic pipetting station (Eppendorf, Hamburg, Germany) and an ELx405 automatic deep well microplate washer (BioTek, Winooski, VT, USA). Results were read using an Elx800 reader (BioTek) and xCheck (Idexx) and ProFILE software (Synbiotics).

### 2.5. Cytometric Analyses

Mononuclear cells were isolated using a Histopaque density gradient (Sigma-Aldrich, Taufkirchen, Germany) with blood and spleen tissue, and a Percoll density gradient with tracheal mucosa (Sigma-Aldrich, Taufkirchen, Germany). The individual stages of lymphocyte isolation from the collected samples were consistent with previously described procedures [16,21,22]. After the isolation steps, the viability and concentration of the lymphocytes were evaluated using a Vi-cell automatic cell counter (Beckman Coulter, Indianapolis, IN, USA). The percentage contents of T lymphocyte (CD4^+^, CD8^+^) and B lymphocyte (IgM^+^) subpopulations were also analyzed in the samples using mono- and polyclonal antibodies (Bio-Rad, Hercules, CA, USA). Cytometric analysis and cell immunophenotyping were performed by means of a FACSCanto II flow cytometer (Becton Dickinson, Franklin Lakes, NJ, USA), FACSDiva Software 6.1.3 (Becton Dickinson, Franklin Lakes, NJ, USA), and FlowJo V10 7.5.5 software (Tree Star Inc., Ashland, OR, USA).

### 2.6. Statistical Analysis

The effect of the deodorizing preparation on the previously noted biochemical parameters and immune system function markers was determined using Student’s t-test for independent groups. The statistical analysis of result was conducted with *Statistica* 13.1 software (StatSoft, Krakow, Poland), and differences were considered significant at *p* ≤ 0.05.

## 3. Results

### 3.1. Biochemical Analyses

The results of the biochemical analyses of serum samples collected from parent-stock chickens, broiler chickens, and broiler turkeys are presented in Table 2. Statistically significant differences were noted in the mean CK activity between the control and experimental groups of parent-stock chickens; in AST (*p* = 0.0045), LDH (*p* = 0.0041), and creatine levels (*p* = 0.02) between respective groups of broiler chickens; and in AST activity between the groups of broiler turkeys.

### 3.2. Serological Analyses

Geometric means of specific antibody titers determined in serum samples are presented in diagrams (Figure 1) plotted for control and experimental groups. Regardless of the vaccine target, no statistically significant differences were reported between antibody titers among the groups of parent-stock chickens, broiler chickens, and turkeys.

### 3.3. Cytometric Analyses

The percentage contents of lymphocyte T and B subpopulations in the samples of tracheal mucosa, blood, and spleen of the birds from the control and experimental groups are presented in graphs (Figure 2). There were no statistically significant differences in the percentage contents of the analyzed subpopulations of immunocompetent cells among all the analyzed samples.

## 4. Discussion

The Deodoric biopreparation merges a microbiological component with a mineral component and was developed to reduce emissions of ammonia and other volatile odorous gases in poultry houses [18]. Gałęcki et al. [10,17] have demonstrated that Deodoric sprinkled on litter where various species and types of poultry were housed had a positive effect on bird welfare principally by reducing ammonia concentration and humidity in poultry houses. Suppression of ammonia production from such operations is essential because high concentrations of odorous gases derived from agriculture and animal production have adverse effects on both the natural environment and living organisms, as confirmed in multiple earlier studies [9,13,23,24,25,26]. The present study continues earlier experiments [10,17] and, emulating the conditions in those, the birds from all experimental groups were kept under the same zoohygienic standards and these were appropriate for the rearing process.

The designed experimental conditions and the high efficiency of the ventilation system enabled the average ammonia concentration to be maintained at 7 to 26 ppm [10,17]. The only difference between the control and experimental groups was the use of the mineral-microbial preparation on the litter. The Deodoric preparation used in the experimental groups’ housing reduced NH_3_ emissions, thus positively contributing to body weight gains of the birds [17].

Previous investigations have shown adverse effects of odorous gases on immune system functions and post-vaccination immunity development. McFarlane and Curtis [27] and McFarlane et al. [28] demonstrated that ammonia concentrations above 125 ppm caused a significant increase in the number of heterophils and a decrease in the numbers of lymphocytes and basophils in birds. Ammonia concentrations at 26–60 ppm contributed to reduced total levels of IgY, IgM, and IgA antibodies and a lower post-vaccination titer of NDV-specific antibodies in birds [26,29]. The serological analysis of serum samples conducted in the present study between control and experimental group of parent-stock chickens, broiler chickens, and broiler turkeys showed no significant differences in post-vaccination immunity development. Likewise, the cytometric analysis demonstrated no significant differences between the analyzed groups of birds. These results indicate that despite the positive impact of the deodorizing preparation on improving zoohygienic conditions, such small differences in NH_3_ concentration between the investigated groups of parent-stock chickens, broiler chickens, and broiler turkeys as well as the relatively low NH_3_ level in the control group (i.e., turkeys—26 ppm, chicken broilers—16 ppm and parent-stock chickens—12 ppm), approximating the permissible level, were insufficient to elicit a modulating effect on immune function. Most of the earlier research works were conducted where levels of harmful gases were very high, with NH_3_ concentration peaking even at 200 ppm [28,30]. In turn, the lack of statistical differences implies no adverse effect of the preparation on the analyzed parameters.

The research findings in the available literature support a correlation between the concentration of NH_3_ and the duration of bird exposure to it and the gas’ effect on immune system performance [23,26,27,29,31]. While no significant differences were found between the control group and the experimental group exposed to a high NH_3_ concentration for one week, 14-day exposure to an NH_3_ dose exceeding 52 ppm resulted in noticeable changes in the immune system [26,29,31]. In the case of parent-stock chickens, which are reared for a longer period than broilers, the likelihood of upsetting body homeostasis by long-term exposure to adverse environmental conditions and harmful gases is substantially higher [25]. Despite such relationships, the present study performed with birds of different production types and in various experimental periods (from 45 to 145 days) showed no significant differences in ammonia effects on their body functions.

Previous findings by Li et al. [25] could only be corroborated by the results of biochemical analyses indicating significant differences in AST, CK, LDH, and creatine levels between the control and experimental groups, which however still fell within the upper recommended levels for individual poultry species [32]. It is worth noting that there were no statistically significant differences in the remaining analyzed biochemical markers and that in all groups examined their values fell within the physiological ranges set for individual bird species [32].

## 5. Conclusions

The presented study results showing the levels of biochemical markers not to be outside the physiological ranges and statistical differences in post-vaccine antibody titers and percentages of T and B lymphocyte subpopulations not to exist imply that there was no adverse effect of the mineral-microbial Deodoric preparation components on the bodies of meat-type breeder chickens, broiler chickens, or broiler turkeys.

## Figures and Tables

**Figure 1 animals-11-02592-f001:**
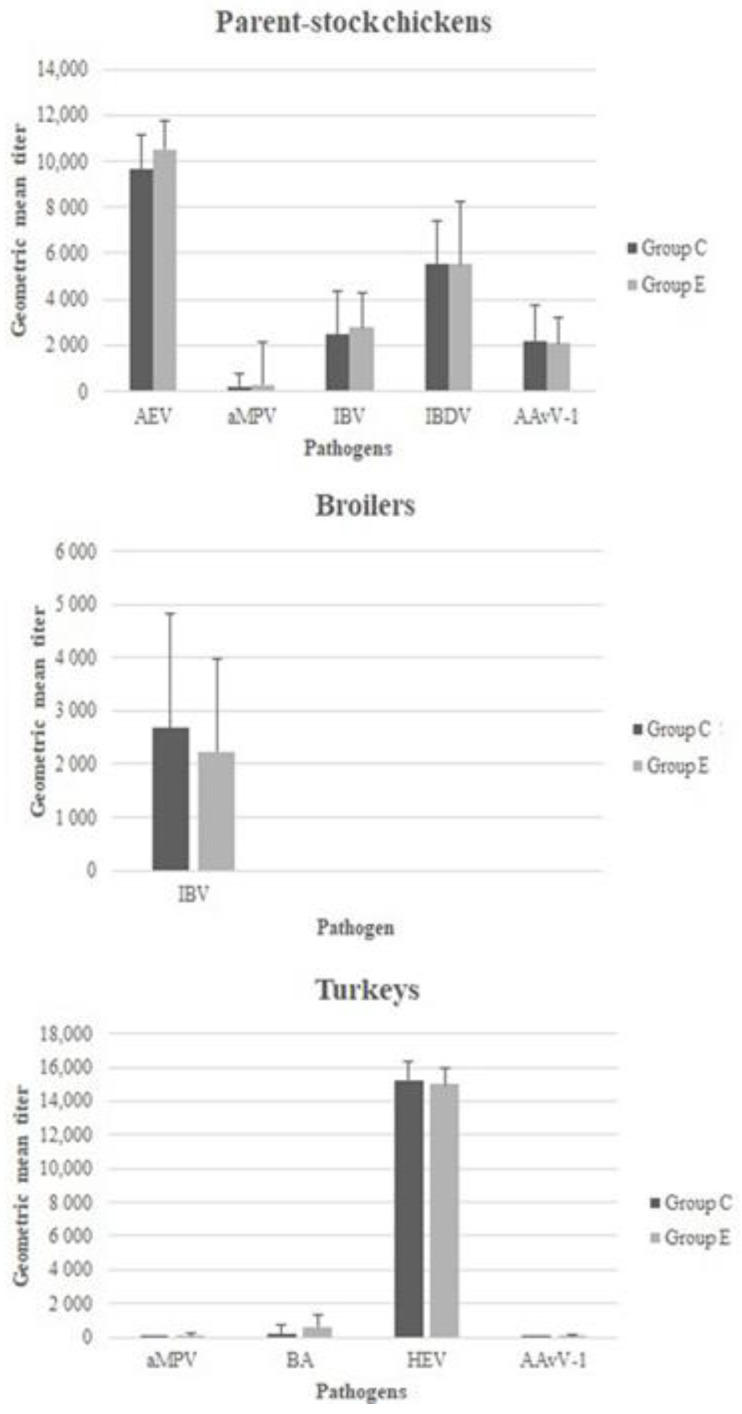
Mean specific-IgY (AEV—Avian encephalomyelitis virus, aMPV—Avian metapneumovirus, BA—Bordetella avium, HEV—Hemorrhagic Enteritis Virus, IBDV—Infectious Bursal Disease Virus, IBV—Infectious Bronchitis virus, AAvV-1—Newcastle Disease Virus) geometric titers in serum samples in two groups (C—control, E—experimental) of parent-stock chickens, broiler chickens, and turkeys.

**Figure 2 animals-11-02592-f002:**
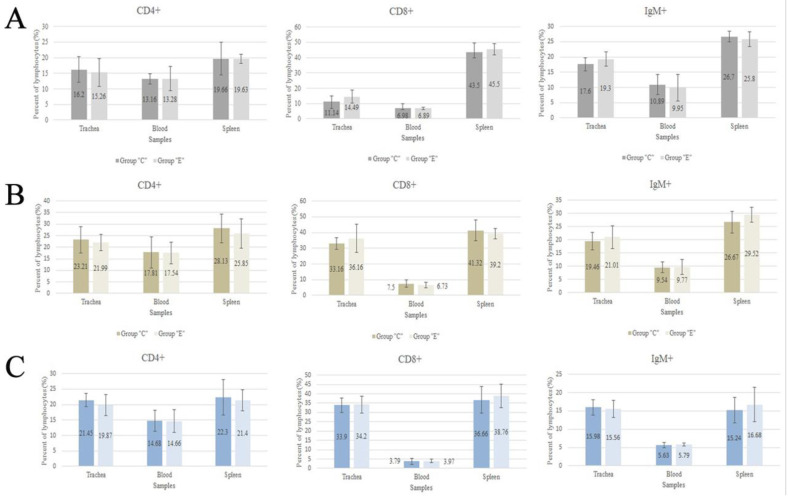
Percentages of T (CD4+ and CD8+) and B (IgM+) cell subpopulations in trachea, blood, and spleen samples in control (group “C”) and experimental group (group “E”) of parent-stock chickens (**A**), broiler chickens (**B**) and turkeys (**C**).

**Table 1 animals-11-02592-t001:** Vaccination schedule applied at the parent-stock chickens flock.

Age *	Disease **	Type of Vaccine	Method of Vaccination
1 d	MD IB	live live	intramuscular injection coarse spray
9 d	*coccidiosis*	live	drinking water
18 d	IB + ND	live	drinking water
28 d	IBD	live	drinking water
50 d	IB + ND	live	drinking water
8 wk	IBD	live	drinking water
9 wk	REO MG *salmonellosis*	live live inactivated	intramuscular injection eye drop intramuscular injection
10 wk	IB	live	drinking water
11 wk	IBD	live	drinking water
12 wk	CIA	live	drinking water
13 wk	AE	live	drinking water
14 wk	IB + ND	live	drinking water
15 wk	REO *salmonellosis*	inactivated inactivated	intramuscular injection intramuscular injection
16 wk	SHS	live	drinking water
19 wk	IB+ND+EDS+SHS IBD + REO	inactivated inactivated	intramuscular injection intramuscular injection

* d = day; wk = week. ** AE—Avian encephalomyelitis; CIA—Chicken infectious anemia; EDS—Egg drop syndrome; IB—Infectious bronchitis; IBD—Infectious bursal disease; MD—Marek’s disease; MG—Mycoplasma gallisepticum; ND—Newcastle disease; REO—reovirosis; SHS—Swollen head syndrome.

**Table 2 animals-11-02592-t002:** Values (±SD) for clinical chemistry parameters in serum samples of parent-stock chickens, boilers, and turkeys *.

Chemistry Parameters	Units	Parent-Stock Chickens		Broilers		Turkeys	
		Control Group	Experimental Group	Control Group	Experimental Group	Control Group	Experimental Group
ALT	U/L	11.5 ± 1.84	11.87 ± 2.57	3.6 ± 0.97	3 ± 1.94	14.75 ± 3.13	14.2 ± 2.35
AST	U/L	252.75 ± 23.24	230.25 ± 45.7	146 ^B^ ± 23.22	175.67 ^A^ ± 14.96	9 ^B^ ± 5.47	19.3 ^A^ ± 27.10
ALP	U/L	188.37 ± 37.95	178.87 ± 56.71	1879.8 ± 539.12	2262.56 ± 655.99	738 ± 215.42	656.5 ± 175.05
Calcium	mmol/L	4.72 ± 0.31	4.52 ± 0.39	2.678 ± 0.12	2.82 ± 0.68	2.66 ± 0.14	2.96 ± 0.96
CK	U/L	3118.18 ^B^ ± 643.54	6001.12 ^A^ ± 1846.39	5139.15 ± 855.51	4003.47 ± 1605.27	5231.88 ± 1852.68	5469.61 ± 1821.56
Creatinine	µmol/L	4.41 ± 1.61	5.85 ± 2.52	4.61 ^B^ ± 0.47	5.79 ^A^ ± 1.37	4.15 ± 1.44	5.79 ± 3.15
LDH	U/L	1044.62 ± 136.7	1141.37 ± 203.06	2191.7 ^A^ ± 535.06	1473.78 ^B^ ± 389.52	2629.87 ± 696.27	2687.4 ± 533.28
Phosphate	mmol/L	2.53 ± 0.54	2.24 ± 0.38	2.41 ± 0.86	2.25 ± 0.54	2.07 ± 0.48	2.27 ± 0.31
Total protein	g/L	51.51 ± 3.65	50.92 ± 2.81	34.27 ± 10.66	32.31 ± 6.68	35.52 ± 82.1	35.25 ± 5.17

* ALT—alanine aminotransferase; AST—aspartate aminotransferase; ALP—alkaline phosphatase; CK—creatine kinase; LDH—lactic dehydrogenase; ^A,B^—Statistically significant difference between the mean value for parameters in groups (control and experimental) (Student *t* test, *p* ≤ 0.05).

## Data Availability

Data is contained within the article. The datasets used and/or analyzed during the current study are available from the corresponding author on reasonable request.

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
