# Peer review of "Effect of a Mineral–Microbial Deodorizing Preparation on the Functions of Internal Organs and the Immune System in Commercial Poultry"

_animals, 2021, doi:10.3390/ani11092592_

Round 1

Reviewer 1 Report

My suggestions, not mandatory, but which I consider important for the authors:
1) If so, include information on the physical, chemical and biological quality of the poultry litter, always obtained before and after the use of additives. The fact is that physical environmental factors (such as bed and air temperature and humidity) are directly related to the generation of ammonia and greenhouse gases.

2) In this case, the temperature of the environment where the birds were kept was kept at 20C, and the relative humidity of the environment was also kept relatively constant. To achieve this control of the environment, the air had to be constantly renewed and/or treated. Under these conditions, ammonia emission and odors will be kept under control at very low levels (less than 20 to 25 ppm in the case of ammonia).
Therefore, the non-difference between treatments is probably due more to good ambient air quality than to the product incorporated into the bed.

Therefore, I suggest the authors use these arguments to help explain the research results.

Reviewer 2 Report

The manuscript entitled “Effect of a mineral–microbial deodorizing preparation on the functions of internal organs and the immune system in commercial poultry” is interesting and well-written. In this respect, the manuscript could be accepted for publication after minor revision.

Major and minor points:

  1. The study was conducted in 2016–2017. Can you write down the date of the experiment in more detail?
  2. In line 115-116, How many chickens were killed? Are they all 120 hens?
  3. In line 135- 136, How many repeats of experimental group of 100 birds?
  4. For the result of Table 2, Each value is the mean, the result is the mean plus and minus the standard deviation or the standard error (mean±SD or mean±SE ) is more precise.
  5. For the result of Figure1 and Figure 2, the columns of each bar chart contain the average value, but there's no standard deviation or standard error.

Reviewer 3 Report

Dear authors, thank you very much for the interesting manuscript. The experimental design is straightforward and suitable for your aims. Unfortunately, the manuscript seems to have been prepared a little careless and in a hurry. The results presented do not mach your Materials and Methods section (did you study breeder chicken Ross 308 or Laying Hens, an if so which strain?). The results also do not match you discussion. You present results on physiological and heath parameters dependent on the Use of deodoric but you discuss the effect of different ammonia concentrations on birds. In the results there are significanat differences for some parameters, but they are not discussed and in the abstract you even say, there were no significant differences at al. Overall too many confusing findings. I must recommend to reject your manuscript, because it ist not clear to which animals your results actually refer to. Additionally, the discussion is insufficient for the results presented.

Please see detailed comments as follows:

Simple summary last line: there are not result for improving housing conditions and reducing gas emissions in this paper.

Abstract L 28: this is not true. In the results section, there are significant differences for some biochemical parameters.

L92 Please explain the term “LOCK”

L95 Please explain “w/v”

L95 Please explain “DE= 7–13, HORTIMEX Sp. z o. o.”

L104 Why was the experiment length of 145 days chosen? Please explain

L 108 Please specify the “obligatory Standards”, maybe refer to the appropriate management guidebook

L 112 The Ross Management Guidebook should be included in the references

L112: Why do you cite your own publication here?

L120 Why didn’t you start the experiment with younger animals?

L125 Please provide more husbandry details, like stocking density, litter and litter renewal, type of food, any treatments/vaccinations? Microclimate,  any relevant management guides?

L126 Why was a higher dose chosen for the turkeys?

L136 Please provide more husbandry details, see above

L138 Why did you use the same dose for young broilers as you used for parent stock and laying hens? Wouldn’t a lower dose be more appropriate for smaller animals? How is this calculated?

L148 are these the samples taken in total or per experiment? Please specify

L195 p not P

Results

L201 Why are there results of laying hens? Laying hens are not mentioned in the Material and Methods section

Table 2: The heading refers to Laying Hens. In the table body there are no results for laying hens, but for breeder chicken. Obviously there seems to be a confusion about the type of animals your results refer to. I cannot verify, whether only the words “Laying hens” and “breeder chicken” are mixed up or whether your results are mixed up, too. Therefore I must recommend a “major revision” – please check carefully, whether you put the correct results to the correct type of bird!

Table 2: a simple referral to the material and methods section is not appropriate as table heading. Please see authors instructions. You need to explain all abbreviations used in the table heading if they occur for the first time.

Table 2: p not P, in the statistics section (L195) you include 0.05, here you don’t. please use a consistent definition for p

L208-210: I don’t understand the meaning of this sentence. Please try to make the meaning more clearly.

Figure 1: Again, you refer to Laying hens, this time in a parenthesis behind breeder chicken. In the material and methods section breeder chicken are specified as Ross 308. Ross 308 are not Laying Hens but Broilers. Therefore it is not clear to what type of bird your results refer to. The definition for “APV” is missing. APV is not mentioned in the material and methods section. Is it the same as “aMPV”? If so, please use consistent terms throughout the paper. “AAvV-1” is mentioned in the material and methods section and in the figure caption, but not in the figure itself. What does group C and E mean? This is not explained in the material and methods section. This part of the results is a little confusing.

Figure 2: The font size is to small. Please amend the axis labelling accordingly. There is no need to repeat “group C and group E” so many times in the figure

L240-241: Does this sentence refer to the present study? Then the citation of your previous studies is not appropriate. Or does it refert to you previous studies? Then it belongs into the literature section and not in the discussion

Discussion

The Discussion of the significant differences of several biochemical parameters is missing.

On the other hand, you discuss results (ammonia), which are not part of this paper. Therefore the discussion needs a major revision.

L256: you did not present results between the groups of birds, but between the experimental and control group of the different types of birds.

L255-260: you didn’t present results for NH3 concentrations. If you want to discuss NH3 results of the same birds with the results presented in this paper, they should be clearly stated in the literature section, if you published them already or you write them in the results section.

L259: Which control group do you refer to? See comment above. This section of the discussion is quite confusing. 26ppm NH3 is rather high. In the German Animal Husbandry order, for example, NH2 concentrations my not exceed 20 ppm for laying hens and broiler chicken.

L274: Which experiment lasted 188 days? In MM section the longest experiment is 145 days for breeder chicken, followed by 88 days for turkeys and 45 days for broiler chicken. Please check this.

L275: You did not present results for the effect of ammonia on the health status of the birds but for the use of deodoric. The discussion does not seem to be focused on the results presented, but on ammonia results which are not part of this paper.

L276: Lee studied the effect of ammonia on laying hens. The results cannot be compared directely, as ou studied other birds and not the effect of ammonia but of deodoric.

Literature:

Citations should be ordered by appearance in the text and not in alphabetical order. Please check authors instructions.

Round 2

Reviewer 3 Report

In my opinion the manuscript is now acceptable for the publication in the present form.

Author Response

The authors are grateful for the Reviewer’s valuable comments and suggestions, which allowed improving manuscript quality.